# Toxicity Mechanisms of Microplastic and Its Effects on Ruminant Production: A Review

**DOI:** 10.3390/biom15040462

**Published:** 2025-03-21

**Authors:** Mengrong Su, Shangquan Gan, Rui Gao, Chunmei Du, Chen Wei, Ali Mujtaba Shah, Jian Ma

**Affiliations:** 1College of Coastal Agricultural Sciences, Guangdong Ocean University, Zhanjiang 524088, China; mengrong.su@gdou.edu.cn (M.S.); gaorui@gdou.edu.cn (R.G.); duchunmei@gdou.edu.cn (C.D.);; 2College of Animal Science and Technology, Northwest A&F University, Xianyang 712100, China

**Keywords:** microplastic, ruminants, toxicity mechanism, oxidative stress, microorganism

## Abstract

Plastic pollution has become one of the major environmental problems facing human beings in the world today. Plastic waste accumulated in the environment forms plastic particles of different sizes due to farming activities, climate change, ultraviolet light, microbial degradation, and animal chewing. The pollution caused by microplastics has become a major environmental problem in recent years, and it is also a research hotspot in the field of ecological environment. More and more studies have found that ruminants are exposed to microplastics for a long time, which seriously threaten their healthy growth. This paper introduces the current situation of plastic pollution; the properties of microplastics and their effects on the ecological environment, human beings, and animals; summarizes the types and toxicity mechanisms of microplastics; and concludes the main ways that microplastics enter ruminants and their harm to them. In addition, the shortcomings and future development of microplastics in ruminants research are summarized and prospected to provide theoretical reference for the related research on alleviating the influence of microplastics on ruminant production.

## 1. Introduction

With the development of industry and the growth of people’s demand for plastic products, the global production chain of plastic-related products has expanded dramatically in the past few decades. Currently, approximately 400 million tons of plastic material is produced globally each year, which is expected to double by 2050 [1]. Due to the widespread use of masks, gloves, and other plastic materials, the Corona Virus Disease 2019 pandemic has aggravated the pollution of plastics on the environment, which not only destroys the ecological environment, but also has a huge impact on animal and human health. Thus, plastic pollution has become one of the most urgent problems facing human beings in the world today [2]. Plastics are widely used in various fields because of their durability, strong corrosion resistance, lightweight properties, and low manufacturing costs. However, the characteristics of plastics that are difficult to degrade, as well as improper recycling, lead to a large amount of plastic waste accumulation in the environment, causing a huge problem for the ecological environment. According to statistics, only 9% of plastic waste in the environment is recycled, and the remaining plastic waste remains in the environment [3]. Plastic recycling significantly reduces energy consumption, fossil fuel usage, and landfilling, thereby decreasing greenhouse gas emissions and lowering the carbon footprint, while contributing to the global economy. Research indicates that recycling 1 ton of plastic can save approximately 130 million kilojoules of energy [4]. Additionally, despite the challenges, plastic recycling has the potential to generate substantial profits of up to USD 60 billion by 2030 within the petrochemicals and plastics sector [5]. Plastic waste that is not recycled forms plastics of different sizes through physical friction and chemical aging, as follows: macro plastics (diameter > 20 mm), medium plastics (5–20 mm), microplastics (MPs) (<5 mm), and nanoplastics (NPs) (<1 μm) [6]. Among them, MPs are classified as emerging and ubiquitous pollutants [7]. MP pollution has been listed as the second key scientific problem in the field of ecology and environment, which researchers have paid more and more attention to and is currently a research hotspot in the field of ecology and environment [8].

MPs refer to solid plastic fragments or particles with a diameter of <5 mm that are made of different polymer monomers and additives [9]. MPs are widely found in soil, drinking water, air, and other environmental media, posing risks to the healthy living environment of humans and animals (Figure 1A). The detection and analysis of MPs face unique challenges, typically involving steps such as sampling, identification, analysis, and qualitative and quantitative assessment, with different detection technologies being employed at each stage. The methods for sampling MPs in the air include active and passive techniques, as well as biological monitoring, the latter of which offers advantages such as low cost, simplicity, extensive spatial coverage, and the ability to reflect the biological effects of pollutant exposure. Identification and analysis techniques include optical microscopy, stereomicroscopy, atomic force microscopy (AFM), and the hot needle method. Qualitative and quantitative techniques include Fourier transform infrared spectroscopy (FTIR), Raman spectroscopy, pyrolysis-gas chromatography-mass spectrometry (py-GCMS), and thermal extraction desorption-gas chromatography-mass spectrometry (MALDI-TOF-MS) [10,11,12]. Depending on the source, MPs can be divided into primary microplastics and secondary microplastics. Primary microplastics are intentionally added to consumer and commercial products such as cosmetics, paints, and detergents. Secondary microplastics are smaller plastic fragments formed by processes such as the degradation of larger polymers by physical, chemical, or biological factors. According to their morphological characteristics, MPs can be divided into fibers, particles, films, fragments, spheres, filaments, and lines [13] (Figure 1B), among which fibers and fragments are the main morphologies of MPs, accounting for 56.6% and 34.4% of total MPs, respectively [14]. Due to their small size, large specific surface area, strong hydrophobicity, and difficulty in biodegradation, MPs can be quickly adsorbed to the surface of plants and soils, which are easily ingested by organisms and accumulate in the body, posing a potential threat to human health [15]. Because they can easily and quickly transfer to different areas of the environment, enter the food chain, absorb harmful chemical pollutants, move across trophic levels, and migrate to different tissues of animals, MPs are generally considered to be more harmful to the environment and human beings than large plastics [16]. Numerous studies have shown that MPs have detrimental effects on all aspects of ecosystems, including the structure and function of microorganisms, plants, and soils. For example, MPs in soil can lead to the disruption of the digestive system, increased mortality, growth inhibition, decreased reproduction function, and reduced immunity [17,18,19]. One relevant study has found that MP exposure can cause damage to the human gastrointestinal tract, inflammation, a reduction in the mucus layer, oxidative damage, and microbial disturbance [20] (Figure 1C). In addition, previous studies have shown that MPs can accumulate in the gastrointestinal tract, kidney, liver, and brain of aquatic organisms and mammals, which can have adverse effects on these organisms [21,22], including reduced appetite, inflammation, oxidative damage, intestinal barrier dysfunction, and energy disturbance [23]. Deng et al. found that polystyrene microplastics (PS MPs) can induce the disturbance of energy and lipid metabolism, as well as oxidative stress (Figure 2D,E). In addition, PS MPs can increase the acetylcholinesterase (AChE) activity and serum levels of threonine, aspartic acid, and taurine, while decreasing the concentration of neurotransmitter precursor phenylalanine; moreover, the changes in these neurotoxic blood biomarkers may contribute to neurotoxicity [24]. Li et al. have shown that polyethylene microplastics (PE MPs) can trigger intestinal inflammation in mice [25]. Jeong et al. found that the monogonont rotifer (*Brachionus koreanus*) exposed to PS MPs showed a reduced growth rate and fertility, a shortened life span, and prolonged reproductive time [26]. Meanwhile, MPs have also influenced the development of animal husbandry [27,28]. According to relevant studies, 50~60% of the foreign matter found in slaughtered livestock is made up of plastic-based materials [29]. Moreover, MPs can also be used as carriers of pollutants such as heavy metals, antibiotics, persistent organic pollutants, and pesticides, which result in potential threats to the environment, as well as animal and human health. MPs such as polyvinyl chloride and polystyrene in seawater can release zinc and copper metals [30,31].

The continued growth of the global population has led to a rise in demand for meat [32]. With the improvement of living standards, the demand for red meat and milk has increased significantly, and ruminants are one of the main sources of these livestock products. The safety of meat products has a direct impact on the health of consumers, such as effects from antibiotics [33] and hormone residues [34] and the reproduction of harmful microorganisms [35] in meat products. Nevertheless, the problems of MPs that contaminate feed and affect the growth and reproduction performance of ruminants cannot be ignored. In recent years, MPs have been found in bovine follicular fluid [36], milk [37], sheep feces [38], beef, and blood [39], which confirms that ruminants are exposed to MPs. A recent study on a dairy farm in Italy showed that all samples of ryegrass hay were contaminated with MPs, which amounted to 39,300 ± 7020 MPs/kg [40]. Another study in India found that 100% of dairy cow diet samples were contaminated with polyethylene terephthalate microplastics (PET MPs), and the content of PET MPs in feed ranged from 89 to 326 g/kg [41]. At present, although the research on the adverse effects of MPs on ruminants is still at an initial stage, it is still necessary to conduct a systematic review of the toxicity mechanism of MPs and their negative effects on ruminants. Firstly, this paper summarized the types of MPs, which can be divided into PE MPs, polypropylene microplastics (PP MPs), and polyvinyl chloride microplastics (PVC MPs), according to the chemical composition. Secondly, the toxicity mechanism of MPs was summarized, including oxidative stress, intestinal damage, and reproductive toxicity. Next, the main ways that MPs enter ruminants and their harm to them were concluded, such as the effects on immune function and rumen microbial community. Finally, the shortcomings in the research on the effects of MPs on ruminants were summarized, and future research was prospected. In recent years, numerous studies have reviewed the biological impacts of MPs on the environment, aquatic systems, and humans. To the best of our knowledge, there has been no previous systematic review specifically addressing the main pathways through which microplastics enter ruminants and their effects on these animals. Filling this gap in the literature represents a significant aspect of our work. Additionally, the purpose of this paper is to enable researchers engaged in relevant fields to further systematically understand the toxicity mechanism of MPs and their harm to ruminants, which can provide a new direction for the development of related fields.

**Figure 1 biomolecules-15-00462-f001:**
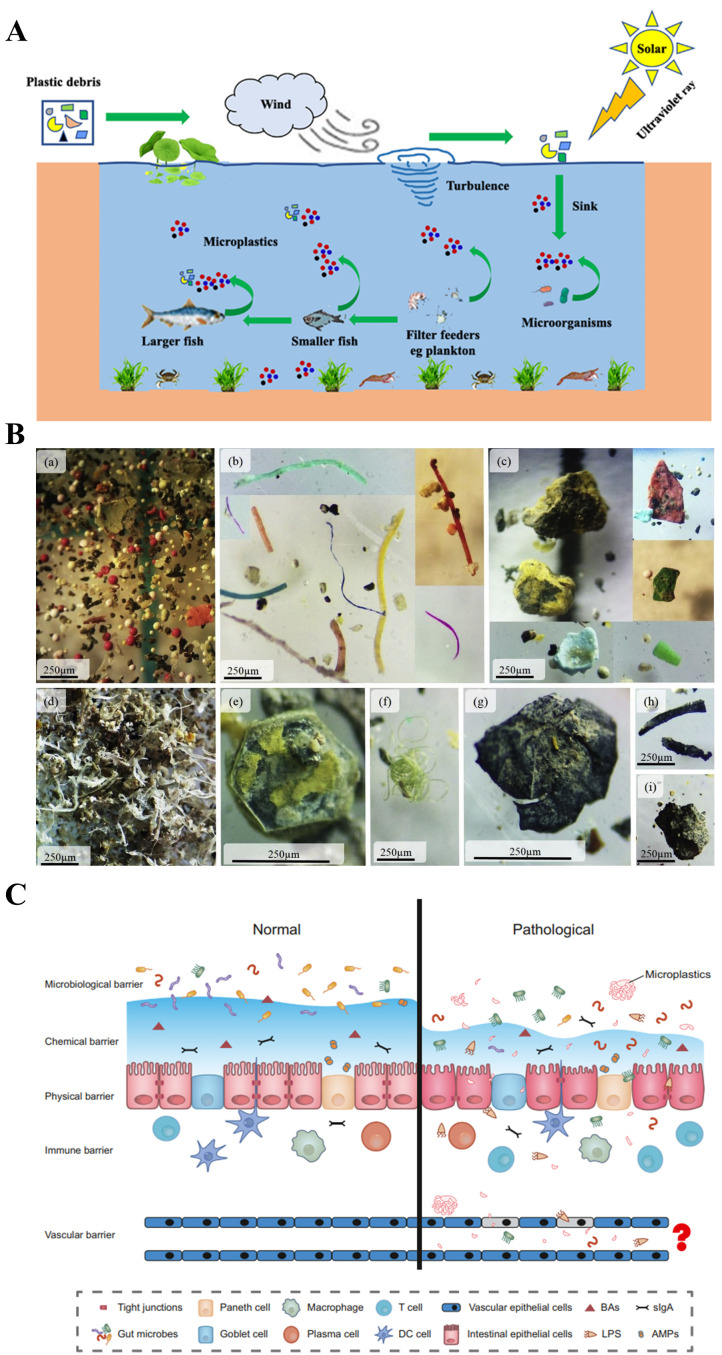
Schematic diagram of the formation process and types of MPs and their effects on the human gut. (**A**) The formation process of MPs [2]. (**B**) Optical microscope images of different types of MPs and microrubbers (MR): (**a**) spherical MPs, (**b**) fibrous MPs, (**c**) fragmented MPs, (**d**) membrane MPs, (**e**) single hexagonal membrane MPs, (**f**) linear MPs, (**g**) membrane MR, (**h**) fibrous MR, and (**i**) fragmented MR [14]. (**C**) In the human gut, MP exposure can lead to an increase or decrease in mucus, changes in the composition and abundance of gut microbes, inflammation of intestinal cells, deficiency of tight junction proteins, and enrichment of immune cells. In addition, MPs may also affect the vascular barrier, but this has not been proven [20].

**Figure 2 biomolecules-15-00462-f002:**
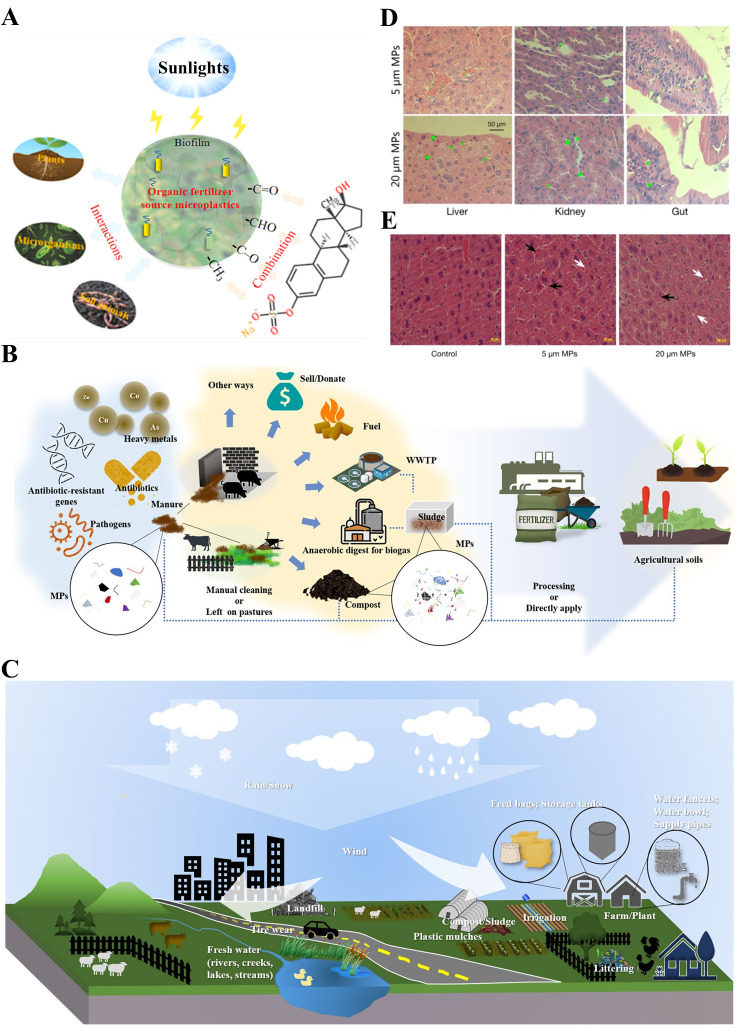
Schematic diagram of MP accumulation in tissues, environmental behavior in soil, and possible sources of MPs in livestock and poultry breeding. (**A**) Environmental behavior of organic fertilizer MPs in soil [42]. (**B**) Possible sources of MPs in livestock breeding [43]. (**C**) The fate of MPs is linked to the disposal of livestock and poultry waste [43]. (**D**) MPs accumulation of different sizes in mouse tissues after 28 days of MPs exposure, and the green fluorescence indicates MPs [24]. (**E**) Representative images of H&E-stained liver sections of mice exposed to 0.5 mg/d PS MPs (5 μm and 20 μm) for 4 weeks (black arrows for liver inflammation, white arrows for fat droplets) [24].

## 2. Methods

Related document information was collected from databases that included Web of Science, Science Direct, PubMed, Scopus, and Google search engines. The key search terms used in various combinations included, but were not limited to, the following: microplastic, toxicity mechanism, ruminants, oxidative stress, rumen microbiota, and immunotoxicity. To exclude unnecessary papers, various filter preferences were employed. The classifications, including “article”, “review”, and “conference paper”, were chosen as the “document types”. The “source type” was selected as “journal” and “conference proceeding”. The search was limited to articles published between “2010 and 2024”, and the selected “language” was “English”. Generated abstracts were screened for eligibility to be included in the review. The search strategy used in this study was adapted to the specific conditions of each database’s search engine.

## 3. Types of Microplastics

According to statistics, approximately 400.3 million tons of plastic were produced globally in 2022 [44]. Among the plastics produced, polyethylene, polystyrene, polypropylene, polyvinyl chloride, polyethylene terephthalate, and polyurethane account for about 92% of total global production [45]. Based on their chemical composition, the MPs can be classified into PS MPs, PE MPs, polypropylene microplastics (PP MPs), PVC MPs, and so on. Table 1 summarizes the main types and risks of MPs. MPs have become ubiquitous pollutants that can be detected in a variety of substrates, including air, water, soil, domestic animals, and wildlife [42,43] (Figure 2A–C). MPs not only seriously damage the ecological environment, but also threaten the life and health of humans and animals, as well as the stable development of animal husbandry.

### 3.1. Polyethylene Microplastics

Polyethylene (PE) is a thermoplastic made of ethylene through addition polymerization. It is the most widely used plastic polymer, and its structure can be expressed as follows: -[CH_2_-CH_2_]_n_- [46,47]. Due to its special chemical structure, PE has the characteristics of low temperature resistance, corrosion resistance, high ductility, chemical stability, and is odorless and non-toxic. PE has been widely used in the production and manufacturing of daily necessities, agriculture, and construction fields, and has become a common material in our daily life [48]. However, due to the large output and low recovery rate, polyethylene is one of the most difficult plastics to degrade, and its accumulation in the environment has caused serious pollution to the ecological environment. For example, PE plastics can be stored in wet soil for 12 to 13 years with only partial degradation and negligible mass loss [49]. PE plastics can be decomposed into PE MPs, with a particle size of less than 5 mm, due to farming activities, climate, solar ultraviolet rays, microbial degradation, and animal chewing [50,51]. Recently, Liu et al. constructed a carp model exposed to PE MPs, and the results showed that MPs can reduce MicroRNA-21 (miR-21) in muscle tissue, negatively regulate interleukin-1 receptor-associated kinase 4 (IRAK4), activate the nuclear factor κB (NF-κB) pathway, induce inflammation, and lead to endoplasmic reticulum stress and apoptosis [52]. Another study found that the ingestion of PE microbeads in mice disrupted the liver’s detoxification response, promoted oxidative imbalances, increased the expression of inflammatory cytokines, and exacerbated liver fibrosis, which demonstrated the adverse effects of PE ingestion on the liver in mammals for the first time [53].

In addition, the extensive use of PE plastics has also seriously restricted the development of animal husbandry. Domestic and wild ruminants often indiscriminately ingest some plastics made of polyethylene, such as licorice nets, whose indigestible nature means that they will accumulate in the rumen after ingestion, resulting in reduced animal health and production performance [54]. In some parts of North America and Europe, the whole corn silage is generally covered in plastic to maintain an anaerobic environment, and, after intake, this plastic will threaten the health of dairy and beef cattle [55]. It is estimated that the diets of ruminants contain 0.07% polyethylene-based plastic [56]. On the other hand, after entering the reticulum of ruminants, these MPs accumulate together to form large globular masses, resulting in the compression of the rumen epithelium during the ruminant process, and, finally, the ruminal function and feed utilization efficiency will be reduced [57,58].

### 3.2. Polypropylene Microplastics

Polypropylene (PP) is a polymer made from propylene through addition polymerization. With the exception of PE plastics, PP plastics are the most widely produced and used plastics at present. The structure of these plastics is similar, and they are difficult to degrade and are characterized by high hydrophobicity and molecular weight. A large amount of PP plastics accumulated in the environment slowly decompose and produce MPs under the action of physics, chemistry, and microorganisms, which not only seriously damage the ecological environment, but also pose a potential threat to the health of humans and other animals. Yang et al. investigated the neurotoxicity and molecular mechanism of co-exposure of bis(2-ethylhexyl) phthalate (DEHP) and PP MPs in immature mice, and the results showed that the co-exposure of these two substances would lead to neurotoxic effects in mice, including the induction of memory deficits and neurocognition, the aggravation of oxidative stress, a reduction in cerebral acetylcholinesterase activity, and damage of hippocampal CA3 ion. Meanwhile, the severity of neurotoxicity in mice gradually increased with the concentration of PP MPs, and the co-exposure of these two substances showed additive or synergistic effects. Additionally, the oral intake of PP MPs can induce colonic cell apoptosis and barrier damage in mice through oxidative stress and toll-like receptor 4/nuclear factor-κB (TLR4/NF-κB) inflammatory signaling pathways [59]. A large quantity of MPs will enter the ocean with wastewater, which not only seriously pollutes the marine ecology, but also has a negative influence on the health and growth of aquatic animals [60]. A previous study found that feeding diets that contained PP MPs for 30 consecutive days had harmful effects on tilapia, including changes in white blood cell spectrum, organ morphology, liver function, and intestinal microbial community, as well as the induction of systemic inflammation [61]. At present, the diffusion of MPs in meat products is an emerging topic, and a relevant study found that 18 different microplastic polymers have been detected in beef burgers, among which polycarbonate, polyethylene, and polypropylene are the most abundant [62].

### 3.3. Polystyrene Microplastics

Polystyrene (PS) is a polymer synthesized from a styrene monomer through free radical polyaddition reaction and is often used in packaging materials, toys, and disposable tableware. When the PS plastic products undergo physical, chemical, and biological processes in the environment, they are degraded and form PS MPs. Due to the characteristics of wide source, easy migration, and small particle size, PS MPs are called “PM_2.5_ in water”. They cause a variety of potential hazards to the ecological environment and biological health [63]. PS MP exposure can cause dysbiosis of the intestinal microbiota [25], induce testicular inflammation and impair the blood–testis barrier in mice [64], and also lead to changes in the intestinal microbiota and tissue metabolism of zebra fish [65]. A recent study has shown that PS MPs can affect lipid metabolism in the macrophages of mice, thereby contributing to the formation of atherosclerotic plaques [66]. Another study found that PS MP contamination increased the incidence of diarrhea and impaired intestinal angiogenesis in weaned piglets through the reactive oxygen species/methyltransferase-like protein 3 (ROS/METTL3) pathway [67]. Relevant animal experiments have shown that MPs may reduce the number of sperm and increase sperm DNA fragmentation, thus leading to reproductive toxicity [64]. Moreover, Zhao et al. used pyrolytic-gas chromatography/mass spectrometry (PY-GC/MS) and laser direct infrared spectroscopy (LD-IR) to detect the distribution and characterization of MPs in human testis and semen, and the results showed that a large amount of PS MPs were detected in the testis, accounting for 67.7%; moreover, PE MPs and PVC MPs were mainly detected in the semen [68].

### 3.4. Polyethylene Terephthalate Microplastics

As a main variety of thermoplastic polyester, its molecular formula of polyethylene terephthalate (PET) is HO-[H_2_C-H_2_C-OOC-Ph-COO]_n_-CH_2_-CH_2_-OH, and its structure is connected by a rigid benzene ring and ester groups at both ends, which can effectively prevent the free rotation of flexible hydrocarbon groups. Therefore, PET has good heat, wear, corrosion resistance and dimensional stability, and it is widely used in electrical equipment, food packaging, and accessory production. At present, the consumption of PET has reached an average of about 40 million tons per year, with a cumulative total of more than 100 million tons [69,70]. A large amount of PET accumulated in the environment is degraded, crushed, and transformed into PET plastic pellets or fragments with a size of less than 5 mm. PET MPs are easily eaten by organisms on land and sea or absorbed as pollutants in the environment, which is a great threat to the life and health of animals and humans. A previous study found that the supplementation of PET MPs at a dose of 4 mg/d in a short period of time reduced the diversity of intestinal microbiota and changed the microbial community in mice [71] (Figure 3A,B). An in vitro study used a digestive model to simulate human colon digestion, and the results showed that the PET MP ingestion altered the composition and diversity of the microbial community in the human colon and reduced the number of total viable bacteria [72] (Figure 3D). Recently, in ruminant production, Tassone et al. studied the effects of PET MPs on the degradation and digestion of mixed hay in the rumen–gastroenteric system, and the results showed that PET MPs reduced the degradation rate and digestibility of crude protein; moreover, the concentration of PET MPs at medium and high levels decreased the degradation rate of neutral detergent fiber in the rumen, indicating that PET MP pollution can affect the digestive tract function of ruminants [73].

### 3.5. Polyvinyl Chloride Microplastics

Polyvinyl chloride (PVC) is a kind of polymer formed by free radical polymerization reactions under the action of light and heat or polymerization of vinyl chloride monomers in peroxide and azo compounds, and its molecular formula is as follows: -[CH_2_-CHCl]_n_-. Due to the high strength and non-flammability, as well as climate change and acid and alkali resistances, PVC is widely used in industry and daily life, accounting for 10% of global plastic production [75,76]. PVC has the characteristics of high stable covalent bonds, high relative molecular weight, and high hydrophobicity, as well as the introduction of chlorine atoms that are difficult to be degraded in the environment and will release harmful chlorinated compounds, resulting in a large accumulation in the environment and causing damage to the ecological environment [77,78]. PVC plastic products are transformed into PVC MPs through various processes, including abrasion, decomposition, and improper disposal. PVC MPs are widely presented in ecosystems such as the atmosphere, water, and soil, which poses a potential threat to ecological balance and biodiversity and also affects the overall physiological function and metabolic processes of organisms. Previously, Chen et al. found that PVC MP exposure reduced intestinal mucus secretion, increased intestinal permeability, and altered the intestinal microbial composition of adult male mice. The health risks of PVC MPs to animals still need more attention [79]. At the same time, the results of Zhuang et al.’s study showed that the hepatotoxicity of adult mice induced by PS and PVC co-exposure was manifested by changes in liver histopathology, the activation of oxidative stress, the release of inflammatory factors, and co-exposure also significantly changed the gut microbial structure and disrupted serum metabolic homeostasis [80]. In aquatic animals, PVC MPs may exert negative effects on the reproductive system of common carp by inhibiting the development of the gonads, affecting the gonads and brain structure, and altering the expression of hypothalamic–pituitary–gonad (HPG)-axis-related genes and steroid hormone levels [81]. PVC MPs can also induce the growth inhibition and oxidative stress in juvenile carp [82]. In addition, after ingestion by ruminants, MPs may accumulate in the gastrointestinal tract, leading to physical blockage and affecting digestion and the absorption of nutrients [28,83]. 

**Table 1 biomolecules-15-00462-t001:** Summary of the main types and risks of MPs.

Types	Chemical Structure of Polymer	Application	Risks	References
PE MPs	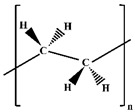	Daily necessities (packaging bags and beverage bottles), agriculture (irrigation hoses and plastic mulch films), and construction fields (paving pads and pipes).	Decreased miR-21 in carp muscle tissue, negatively regulated IRAK4, activated NF-κB pathway, induced inflammation, and led to endoplasmic reticulum stress and apoptosis. Disturbed the detoxification response of mouse liver, promoted oxidation imbalance, increased the expression of inflammatory foci and cytokines, and aggravated liver fibrosis. Reduced the rumen function and the feed utilization efficiency.	[46,47,48,52,53,57,58]
PP MPs	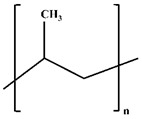	Fertilizer woven bags, auto parts, injector, infusion bottles, and housing for appliances (e.g., washing machines and refrigerators).	In immature mice, it induced memory deficit and neurocognition, increased oxidative stress, decreased acetylcholinesterase activity in the brain, and caused CA3 ion damage in the hippocampus. Oral intake of PP MPs can induce colon cell apoptosis and intestinal barrier damage in mice through oxidative stress and TLR4/NF-κB inflammatory signaling pathways. The changes in tilapia’s white blood cell spectrum, organ morphology, liver function, intestinal microbiota, and systemic inflammatory disorders.	[60,61]
PS MPs	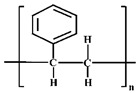	Packing materials, toys, disposable tableware, and bubble wrap.	It can cause the disorder of intestinal flora in mice, induce testicular inflammation, and destroy the blood–testis barrier. The intestinal microbiota and intestinal tissue metabolism of zebrafish were changed. It affected the lipid metabolism of mouse macrophages and then caused the formation of atherosclerotic plaque. Increased the incidence of diarrhea and impaired intestinal angiogenesis in weaned piglets. It may reduce sperm count and increase sperm DNA fragmentation, which can lead to reproductive toxicity.	[25,64,65,66,67,68]
PET MPs	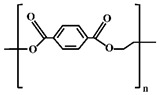	Mineral water bottles, culture plates, plastic test tubes, insulating materials, clothing, and home textiles.	It can reduce the diversity of intestinal microbiota and change the microbial community in mice. In vitro studies of digestive models have shown that PET can alter the composition and diversity of microbial communities in the human colon and reduce the number of total viable bacteria. The degradation rate and digestibility of crude protein in ruminants decreased.	[71,72,74]
PVC MPs	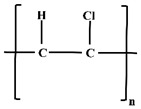	Floors, window frames, sewer lines, and scutcheon.	It increased intestinal permeability, caused intestinal mucus secretion dysfunction, and changed intestinal microbial composition and metabolomic profile of mice. In adult mice, it can change liver histopathology, activate oxidative stress, and release inflammatory factors. In addition, the intestinal microbial structure also was changed. It can inhibit gonad development in carp, affect gonadal structure, and alter the expression of HPG-axis-related genes and steroid hormone levels. Induced growth inhibition and oxidative stress in juvenile carp.	[79,80,81,82]

## 4. Toxicity Mechanism of Microplastics

The toxicity of MPs mainly depends on their type, exposure dose, size, and shape. Toxicological studies on MPs were originally conducted in aquatic organisms, and current studies are mainly from various cellular and rodent models, which are usually studies of toxicological effects observed at high concentrations of polystyrene over short periods of time. However, toxicity studies of MPs on some terrestrial mammals (e.g., pigs, cattle, and goats) are often neglected, and the health of these animals is closely related to the survival of humans. MPs not only cause harm to animal bodies, but the additives used in the plastic manufacturing process and the pollutants absorbed from the external environment can also cause varying degrees of damage to the animals. As shown in Table 2, MPs exert a variety of toxic effects in animals, including oxidative stress, intestinal damage, immunotoxicity, and reproductive toxicity.

### 4.1. Oxidative Stress

A previous study has shown that oxidative stress and membrane damage were important mechanisms of MP toxicity [84]. When the body is exposed to MPs, the production of reactive oxygen species (ROS) will increase, and the activity of antioxidant enzymes will decrease, which then induces oxidative stress. The causes of oxidative stress induced by MPs include the inflammatory response caused by MPs, the increased release of ROS, the presence of metal oxides in MPs, and the adsorption of ambient pollutants. The increased production of ROS is due to the fact that the harm caused by MPs is mainly caused by damaging NADPH oxidase (NOXs), oxygen burst, or the mitochondrial electron transport chain (ETC) [85]. Meanwhile, the strong oxidation of ROS can lead to protein inactivation, lipid peroxidation, and DNA damage. After entering cells, MPs not only induce oxidative stress, but also damage the plasma membrane and induce apoptosis, which seriously affects cell growth [73] (Figure 3C). The exposure of polyvinyl chloride to human blood lymphocytes leads to the formation of large amounts of ROS, lipid peroxidation, and glutathione depletion [86]. Recently, Yaqoob et al. aimed to determine the alleviating effects of rhamnetin (RHM) on liver injury induced by PS MPs in rats. The results showed that PS MP (0.01 mg/kg) treatment decreased the expression of nuclear factor erythroid-2-related factor 2 (Nrf-2) and antioxidant genes and increased the expression of kelch-like ECH-associated protein 1 (Keap-1). In addition, PS MP treatment also decreased the activities of glutathione reductase (GSR), glutathione peroxidase (GSH-Px), superoxide dismutase (SOD), catalase (CAT), and glutathione (GSH) and increased the levels of Malondialdehyde (MDA), ROS, and inflammatory factors; however, the results of the administration group of PS MPs (0.01 mg/kg) + RHM (50 mg/kg) proved that RHM had the effects of liver protection, anti-inflammation, anti-apoptosis, and anti-oxidation [87]. Another study verified that exposure to PS MPs inhibited energy and lipid metabolism, induced oxidative stress, and caused neurotoxicity of skeletal muscle in chicks [88]. MPs also have potential negative effects on freshwater fish. Related research found that fishes exposed to MPs would produce oxidative stress, microbial imbalance, and metabolic changes, which reduced their survival rate [89,90] (Figure 4A).

Deoxyribonucleic acid (DNA) is the most basic genetic material. It is a macromolecular polymer that carries biological genetic information and ensures its normal structure and function, which are extremely important for maintaining body stability. Various types of DNA damage will occur when cells are subjected to various stresses, such as DNA replication errors and base pair changes. One relevant study has demonstrated that MP exposure can cause changes in the cell cycle, resulting in toxic effects such as DNA damage [91]. The main cause of MP damage to DNA is its ability to produce excessive ROS and then cause breakage of DNA strands. A previous study found that MP exposure induced DNA damage in the nucleus and mitochondria, as well as hepatotoxicity and liver fibrosis, of male mice, and, in addition, the mitochondrial DNA was released into the cytoplasm [92]. In mussels, a previous experiment has proved that MPs can cause DNA damage [93], and combining MPs and the plasticizer di (2-ethylhexyl) phthalate (DEPH) can cause cell DNA damage. Wu et al. found that the co-exposure of PS MPs and DEPH can lead to ovarian damage in mice, and both of them can trigger CNR1/CRBN/YY1/CYP2E1 signal axis and promote the overproduction of ROS, causing oxidative stress and eventually resulting in oxidative damage of DNA [94]. Different particle sizes of MPs may lead to different effects and mechanisms of DNA damage. However, existing studies have not yet clarified the specific mechanism of action of MPs on DNA damage. In the future, in vivo and in vitro experiments should be conducted to explore the action pathway of MPs with different particle sizes to cause DNA damage.

### 4.2. Immunotoxicity

Previous experiments on mammalian animals and in vitro cells have shown that MPs ingested into the body are difficult to absorb, and only a small part can enter blood circulation through the lymphatic system [95]. MP particles larger than 0.2 μm in lymph fluid can be removed from the body by the spleen filtration system, but it is unclear whether MPs particles larger than 1.5 μm can be removed [96]. MPs with certain characteristics can be transferred across living cells to the lymphatic and circulatory systems, where they can accumulate in organs and affect cellular health and the immune system [97]. After entering the circulation of organisms, MPs can disturb the immune system and produce immunotoxicity through metastasis and enrichment between tissues and organs, including the ROS-induced inflammatory response, the release of inflammatory factors, and the regulation of the expression of immune genes [98] (Figure 4B). After phagotrophy by phagocytes as a foreign body, MPs can induce granulocyte peroxidation and stimulate immune cells to secrete inflammatory factors, thus increasing the proportion of neutrophils in the blood and immunoglobulin A (IgA) level. Park et al. found that, compared with the control group, the mice that were fed PE MPs for 90 d had an increased proportion of neutrophils, as well as IgA level, in the blood; in addition, the persistence of PE MPs and their migration to the mast cell membrane were observed in the stomach and spleen of mice [99]. On the other hand, some studies have confirmed that MPs have a certain degree of immunocytotoxicity to human cell lines. The results of Hwang et al.’s study showed that 25 μm PP MP exposure at the dosage of 1000 μg/mL induced ROS production, which led to cytotoxicity. Also, MPs can cause the secretion of pro-inflammatory cytokines tumor necrosis factor-α (TNF-α) and interleukin-6 (IL-6) in human peripheral blood mononuclear cells (PBMCs), and increase the release of histamine from mast cell lines [100].

In addition, MP exposure in animals also leads to the production of effector T cells, the low activation of dendritic cells, and the inhibition of helper T cells, resulting in immunosuppressive effects. Previously, an experiment demonstrated that MPs in drinking water significantly impaired the immune function of mice by reducing spleen weight and the number of CD8^+^T cells and increasing the ratio of CD4^+^/CD8^+^T cells. MPs can also induce immune and spleen damage by reducing S100A8 levels and related pathways [101]. Another study found that PE MP exposure increased the intestinal microbial species and diversity in mice and decreased the percentage of Th17 and Treg cells in CD4^+^T cells, resulting in immunosuppression [25].

### 4.3. Reproductive Toxicity

Related studies have verified that MPs also have reproductive toxicity, which has a huge impact on the survival and reproduction of marine and terrestrial biological populations. MPs contain some monomers and additives such as phthalates (PAEs), nonylphenol, and bisphenol A (BPA), and they can interfere with the normal function of the endocrine system in organisms, which are referred to as environmental endocrine disruptors [102,103]. BPA, a xenoestrogen, exhibits hormone-like properties and mimics the function of estrogen, which may be associated with the onset of obesity and can lead to hormone secretion disturbances, as well as the occurrence of cardiovascular disease. Deng et al. have demonstrated that MPs can transport PAE into mouse tissues, and, compared with untreated MPs and PAE alone, MPs contaminated with PAE can induce increased reproductive toxicity, which was mainly manifested as spermatogenesis disorder, physiological changes in sperm, and increased oxidative stress. These findings highlight the potential reproductive toxicity of MP co-exposure with plastic additives in male terrestrial mammals [104]. Additionally, another study found that co-exposure of PS MPs and Bis (2-ethylhexyl) phthalate (DEHP) resulted in cystic and atretic follicles and oxidative stress in the ovaries of rats and further confirmed that co-exposure can activate the tgf-β/Smad3 signaling pathway, whereas the inhibition of this pathway can effectively reduce hormone dysregulation, oxidative stress, and ovarian fibrosis [105]. Even if the concentration of these environmental endocrine disruptors is very low, they will interfere with the physiological activities of endogenous hormones in the body. Relevant studies have found that fish exposure to MPs interferes with the hypothalamic–pituitary–gonadal axis, thereby inhibiting the secretion of sex hormones and decreasing the fertility of adult fish and the growth rate of young fish [106].

At present, some studies have explored the toxic mechanism of MP damage to germ cells, including the involvement of mitogen-activated protein kinase (MAPK) signaling pathway, oxidative stress, and inflammation [107] (Figure 4C). The reproductive toxicity of MPs includes a decline in sperm number and activity, an elevation in sperm malformation rate, a reduction in reproductive ability and testosterone level, impaired integrity of the blood–testicular barrier and decreased offspring survival rate [108]. Xie et al. studied the effects of PS MPs on the reproductive system of male mice, and the results showed that MP exposure resulted in a significant decrease in sperm count and motility and a significant increase in the rate of sperm malformation, as well as a decrease in the activity of sperm-metabolity-related enzymes such as succinate dehydrogenase (SDH) and lactate dehydrogenase (LDH), suggesting that MPs may have adverse effects on the reproductive system by interfering with lipid metabolism and affecting the activity of metabolity-related enzymes. Moreover, MP exposure can cause oxidative stress and activate c-Jun N-terminal kinase (JNK) and p38 mitogen-activated protein kinase (p38 MAPK). When SB203580, a specific inhibitor of p38 MAPK, inhibits the p38 MAPK, MPS-induced sperm damage is reduced and testosterone secretion is improved. Therefore, MPs can induce reproductive toxicity in mice through oxidative stress and the activation of the p38 MAPK signaling pathway [109]. The findings of Li et al. proved that MPs can cause oxidative stress, activate the p38 MAPK pathway, and reduce the nuclear Nrf2 pathway, all of which affect the quantity and quality of sperm and the integrity of the blood–testicular barrier [110].

### 4.4. Neurotoxicity

MPs have become a widespread environmental pollutant, which may be associated with an increased incidence of neurodegenerative diseases. A large number of studies have found that MPs with small particles can enter and accumulate in the brain tissue through the blood–brain barrier for a long time, which will inhibit the expression and transcription of genes related to signal transduction, neurodevelopment, and the activity of enzymes related to neuron and neurotransmitter synthesis, thus affecting the release of neurotransmitters and leading to neurotoxicity [111,112]. The alteration of neurotransmitter release levels is considered a key mechanism driving behavioral changes after MP exposure [113]. A recent study evaluated the effects of aging MPs in plastic bowls (PB) on the neurotoxicity of Caenorhabditis elegans and found that exposure to 0.1–1 mg/L aging plastic bowls (A-PB) had greater neurotoxicity on motor behavior than pure PB without photoaging. Nematodes exposed to the latter showed head twitching and body bending, as well as decreased wavelength and mean amplitude. In addition, the levels of dopamine, serotonin, and GABA were decreased, and the expression of neurotransmitter-related genes (e.g., dat-1, tph-1, and unc-47) also decreased significantly. These neurotransmitter-related genes showed significant positive correlation with motor behavior and played a key role in mediating neurotoxicity of Caenorhabditis elegans [114]. The effects of MPs on neurotoxicity of aquatic organisms have also been studied. Choi et al. found that polyamide (PA) exposure would produce neurotoxicity to juvenile crucian carp, which was mainly manifested in the significant inhibition of AChE activity in liver, gill, and intestine by PA exposure in neurotransmitters [115]. Moreover, relevant research found that MPs caused neurotoxicity by inhibiting AChE, increasing lipid oxidation (LPO) in brain and muscle and changing the activities of energy-related enzymes, including lactate dehydrogenase (LDH) and isocitrate dehydrogenase (IDH) [23].

When oxidative stress occurs in the body, the lipids in the brain are vulnerable to ROS attack and further oxidized to MDA, which will cause structural damage to nerve cell membranes and organelles (e.g., endoplasmic reticulum and mitochondria) and eventually lead to neurological dysfunction [112,116,117] (Figure 4D). A previous study found that, after MP exposure, the brain cells T98G will not show cell lysis, but ROS production will increase, which can cause oxidative stress and neurocytotoxicity. Therefore, oxidative stress may be one of the important mechanisms of MPs to induce cytotoxicity [118]. The results from Umamaheswari et al.’s experiment showed that exposure of healthy fish to different concentrations of PS MPs (10 and 100 μm/gL) increased the ROS levels in the brain and disrupted the antioxidant defense system, and this oxidative stress can lead to nerve damage and is associated with some psychiatric disorders such as schizophrenia, depression, and attention disorders. In addition, relieving oxidative stress in the brain can effectively reduce depressive behavior and protect neurons [119].

**Figure 4 biomolecules-15-00462-f004:**
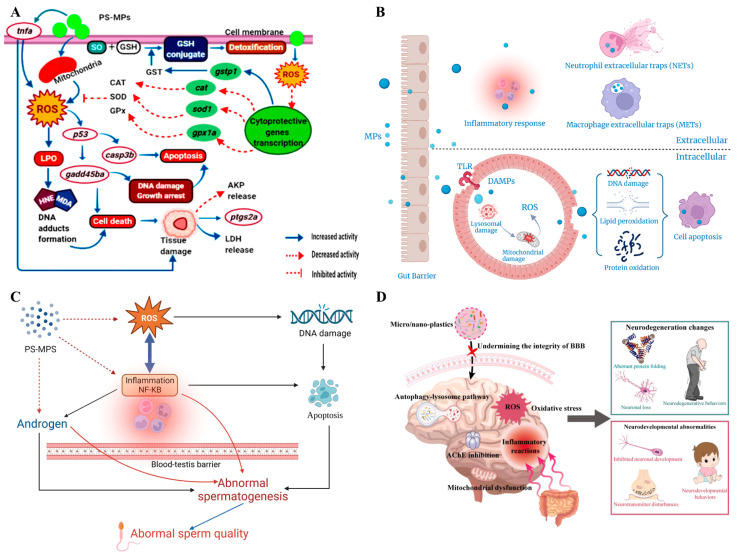
Schematic of cytotoxicity, immunotoxicity, and neurotoxicity of MPs. (**A**) Mechanism of PS MPs induced apoptosis in zebrafish: In general, PS MPs or the unbound styrene (which forms SO (styrene oxide) later) enter the cell and are detoxified by the action of cytp450, GSH, and GST (Glutathione S-Transferase) enzymes. This mitochondrial detoxification results in ROS generation, which stimulates LPO with the release of MDA and HNE. This leads to the formation of DNA adducts causing cell death. Furthermore, the produced ROS upregulates the transcription of the p53 gene, which in turn switches on the casp3b activation. The activated casp3b promotes the transcription of gadd45ba, resulting in DNA damage and apoptosis. This results in tissue damage, causing increased expression of ptgs2a and tnfa with the release of LDH enzymes. On the other hand, the produced ROS downregulates the expression of cytoprotective genes such as CAT, SOD1, and gpx1a and upregulates the expression of detoxification gene gstp1, thereby influencing the translation of antioxidants [90]. (**B**) Potential biological mechanisms of cytotoxicity and immunotoxicity of MPs [98]. (**C**) Abnormal effects of MPs on spermatogenesis in fishes. MPs can increase the standards of the inflammatory cytokines IL-6, TNF-α, and IL-1β in zebrafish embryos. ROS can cause DNA damage in cells, and the factors that repair DNA damage can promote the development of inflammation. Failure to repair DNA damage can lead to cell apoptosis. IL-1β and TNF-α are able to affect hormone synthetase and hormone levels. IL-1β and TNF-α can influence the position and quantity of tight junction proteins through the NF-kB pathway to affect the completeness of the blood–testis barrier. Proper binding of the right amount of androgen and hormone receptor AR has a significant impact on maintaining the integrity of the testis barrier [107]. (**D**) Risks of neurological diseases induced by MPs/NPs and possible mechanisms [112].

### 4.5. Intestinal Injury

Most MPs are ingested orally by animals, thus, they mainly accumulate in the gastrointestinal tract. As the main digestive and absorption organ of animals, as well as the largest immune organ, intestinal mucosa is the first line of defense against intestinal infection and can effectively prevent endogenous and exogenous antigens from entering the systemic circulation through the intestine, which protects the health of the body. A large amount of ingested MPs accumulate in the gastrointestinal tract, which is, therefore, the main target organ for the toxic effects of MPs. At present, many studies have shown that MPs can cause intestinal damage in animals, mainly including reduced intestinal mucus secretion, damage of intestinal mucosa, intestinal inflammation, intestinal barrier dysfunction (e.g., physical, chemical, microbial, and immune disorders), and intestinal microbial disturbance [25,73]. After ingestion, the PE MPs can cause physical intestinal damage of purple mussels [120]. Previously, an experiment verified that feeding 50 μg/d PS MPs significantly aggravated radioactive intestinal injury in mice, which was characterized by a decrease in villi height and goblet cells and a disturbance in the microbial community [121]. In addition, Li et al. found that exposure to high concentrations of MPs (600 μg/d PE MPs) for 5 weeks caused a significant inflammatory response in the intestines of mice, while the expressions of TLR4, Activator protein 1 (AP-1), and interferon regulatory factor 5 (IRF5) were significantly up-regulated [25].

The gut microbiota is critical to the health of the host, as it plays an important role in the digestion and absorption of nutrients, the movement and secretion of the gut, the maintenance of normal development and activity of the immune system, and the integrity of the intestinal epithelial barrier. The effect of MPs on intestinal microbiota has been a hot topic for many researchers at present, as they can affect the diversity and composition of gut microbiota. A previous study has shown that PS MPs not only reduced the secretion of intestinal mucus and impaired intestinal mucus function, but also changed the composition of the intestinal microbiota of mice, which were manifested by a significant decrease in 13 bacterial species (Parabacteroides, Prevotella, Dehalobacterium, Turicibacter, Bifidobacterium, Phascolarctobacterium, Lachnospira, Haemophilus, Adlercreutzia, Megamonas, Blautia, Dialister, and Veillonella) and a significant increase in 2 bacterial species (Coprococcus and Anaeroplasma), and, in addition, PS MPs had a significant impact on the main metabolic pathways of functional genes in the microbial community, leading to changes in the diversity of intestinal microbiota and metabolic disorders in mice [122]. The changes in the gut microbiota are associated with a range of chronic diseases of other organs, such as the cardiovascular system and kidney, and neurological diseases. Also, MPs can cause cytotoxic effects by disrupting the mitochondrial membrane potential and oxidative stress, thus damaging human intestinal cells [123].

### 4.6. Inhibition of Growth and Development

MPs can inhibit the growth and development of animals, including growth retardation, decreased reproductive rate, and shortened life span. PS MPs (40~1300 μm) can significantly inhibit the growth of Arenicola marina (L.), and the degree of inhibition is positively correlated with the concentration of MPs [124]. On the one hand, MPs accumulated in the digestive tract of animals after being ingested can cause physical damage, block the digestive tract, and result in a sense of satiety, which then decreases feeding capacity and hinders the digestion and absorption of dietary nutrients, finally leading to malnutrition and weakening the growth and reproduction ability of animals [125]. A previous study found that the mixture of PS MP and liver homogenate exposure caused a decrease in the metasoma and germ layer area of planaria, indicating that the growth and regeneration of planaria were delayed. In addition, the proliferation and differentiation of stem cells were also inhibited, and the proportion of mitotic stem cells was reduced [126]. Yin et al. found that, compared with the control group, the Sebastes schlegelii in the PS MP treatment group showed decreased food sensitivity and prolonged foraging time, indicating that PS MPs significantly weakened the feeding activity [127]. On the other hand, MPs can lead to the inhibition of the growth of the organism by disrupting the overall homeostasis. A previous study verified that exposure to PE MPs with a particle size of 1~50 μm for 18 days disrupted the overall homeostasis of mussels and caused the production of stress- and immune-related proteins, which can decrease the energy level allocated to growth, ultimately leading to an increase in energy expenditure and a decrease in the growth rate of mussels [128]. In addition, MPs enter the nucleus and cause DNA damage, which can also result in developmental retardation. Nevertheless, the information on the ability of the toxicity mechanism of MPs to inhibit the growth and development of animals is limited. Most of the studies are on aquatic animals, and the toxicity mechanism of MPs on mammals still needs to be further investigated.

**Table 2 biomolecules-15-00462-t002:** Summary of the toxic effects and mechanisms of MPs.

Toxic Mechanisms	Test Subjects	Type and Size	Dose and Exposure	Effects	References
Oxidative Stress	Human blood lymphocytes	PVC MPs (0.16~1.82 μm)	24, 48, and 96 μg/mL (3 h)	ROS mass formation, lipid peroxidation, and glutathione depletion	[86]
	Rat	PS MPs (100 nm)	0.01 mg/kg (30 d)	The expression of Nrf-2 and antioxidant genes was decreased, and the expression of Keap-1 was increased. The activities of GSR, GSH-Px SOD, CAT, and GSH were decreased, and the levels of MDA, ROS, and inflammation were increased.	[87]
	Chick	PS MPs (0.5, 5, and 50 μm)	0.5 mg/mL (35 d)	The smaller the diameter of PS MPs, the more they are deposited in skeletal muscle, and their exposure inhibited energy and lipid metabolism and induced oxidative stress and had the potential for skeletal muscle neurotoxicity.	[88]
	Grass carp	PS MPs (32~40 μm)	100 and 1000 μg/L (21 d)	With the increase in MP concentration, the activity of antioxidant enzymes decreased and the mortality increased.	[89]
	Male mouse	MPs (0.1 and 1 μm)	1 mg/L (24 h)	It can induce DNA damage in the nucleus and mitochondria and cause hepatotoxicity and liver fibrosis, and the content of ATP in liver tissue decreased.	[92]
	Mouse model	PS MPs (5~10 μm)	100 mg/L PS MPs and 200 mg/kg DEHP (35 d)	Joint exposure can lead to ovarian damage in mice. Further research on the mechanism of ovarian granulosa cells cultured in vitro found that co-exposure had a synergistic effect, which can trigger the CNR1/CRBN/YY1/CYP2E1 signal axis, promote the excessive production of ROS, and cause oxidative stress, finally leading to oxidative DNA damage.	[94]
Immunotoxicity	Mouse	PE MPs (40~48 μm)	0.125, 0.5, and 2 mg/d (90 d)	The proportion of neutrophils in the blood increased significantly, and the persistence of PE MPs was observed in the stomach and spleen of the mice. The level of IgA in the blood of the mice was significantly increased and positively correlated with the dose of PE MPs.	[99]
	Human and mouse cell lines	PP MPs (~20 and 25~200 μm)	10, 50, 100, 500, and 1000 μg/mL (48 h)	Exposure of 25 μm PP MPs to 1000 μg/mL induced ROS production, which led to cytotoxicity. The secretion of TNF-α and IL-6 in human PBMCs and histamine release from mast cell lines were increased.	[100]
	Mouse	PS microbeads (5 μm)	5 μg/mL (28 d)	By reducing spleen weight and the number of CD8^+^T cells and increasing the proportion of CD4^+^/CD8^+^T cells, the immune function of mice was significantly impaired. MPs may also induce immune and spleen damage by decreasing S100A8 levels.	[101]
	Mouse	PE MPs (10~150 μm)	2, 20, and 200 μg/g (35 d)	It can increase gut microbial species, diversity of microbiota, and abundance of bacteria and decrease the percentage of Th17 and Treg cells in CD4^+^T cells, resulting in immunosuppression.	[25]
Reproductive Toxicity	Mouse	PE MPs (0.4~5 μm)	0.2 g/L (30 d)	Combined exposure of MPs and plastic additives had potential reproductive toxicity in male terrestrial mammals, which were reflected in spermatogenesis disorder, physiological changes in spermatozoa, and aggravation in oxidative stress.	[104]
	Rat	PS MPs (0.2~0.5 μm)	5 mg/kg/d (42 d)	PS MPs can lead to cystic and atretic follicles and oxidative stress in the ovaries of rats, which further confirmed that co-exposure of DEHP and PS MPs can activate the tgf-β/Smad3 signaling pathway, and the inhibition of this pathway can effectively reduce hormone imbalance, oxidative stress, and ovarian fibrosis.	[105]
	Oryzias melastigma	PS MPs (10 μm)	2, 20, and 200 μg/L (60 d)	It can delay the gonad maturation of female fish and reduce fecundity. The HPG axis was negatively regulated. The transcription of genes associated with the estrogen production pathway was also down-regulated, resulting in lower plasma 17b-estradiol and testosterone concentrations.	[106]
	Mouse	PS MPs (5.0~5.9 μm)	0.01, 0.1, and 1 mg/d (42 d)	It interferes with lipid metabolism, affects metabolism-related enzyme activity, adversely affects the reproductive system, causes oxidative stress, and activates JNK and p38 MAPK.	[109]
	Rat	PS MPs (0.5 μm)	0.015, 0.15, and 1.5 mg/d (90 d)	It can induce oxidative stress, activate the p38 MAPK pathway, and reduce the nuclear Nrf2 pathway, thus affecting the quantity and quality of sperm and the integrity of the blood–testicular barrier.	[110]
Neurotoxicity	Juvenile crucian carp	PA MPs	4, 8, 16, 32, and 64 mg/L (14 d)	The AChE activity of the liver, gill, and intestine was significantly inhibited by PA MP exposure.	[115]
	European perch	Fluorescence red polymer microspheres (1~5 μm)	0.26 and 0.69 mg/L (96 h)	It can inhibit AChE, increase LPO in brain and muscle, and alter the activity of energy-related enzymes, including LDH and IDH, thus causing neurotoxicity.	[23]
	Adult zebrafish model	PS MPs (0.10~0.12 μm)	10 and 100 μg/L (35 d)	MP exposure can induce the production of ROS and destroy the antioxidant defense system, leading to nerve damage, and it has been linked to schizophrenia, depression, attention disorders, and other mental disorders.	[119]
Intestinal Injury	Mouse	PS MPs (5 μm)	50 μg/d (17 d)	RIII was aggravated in mice with TAI, which was characterized by reduced villi height and cupped cells, as well as microbial community disturbance.	[121]
Inhibition of growth and development	Turbellarian	PS MPs (0.1, 1, and 10 μm)	10, 50, and 100 mg/mL (21 d)	The growth and regeneration of planaria were delayed due to the decrease in posterior body and germ layer area. The process of stem cell proliferation and differentiation were also inhibited, and the proportion of mitotic stem cells was reduced.	[126]
	Sebastes schlegelii	PS MPs (15 μm)	1 × 10^6^ microspheres/L (21 d)	It can weaken the feeding activity, affect the energy reserves, and reduce the nutritional quality of the organism.	[127]
	Mussel	PE MPs (1~50 μm)	0.02, 0.04, 0.06, 0.08, and 1.0 mg/L (18 d)	It can disrupt the overall homeostasia in the mussel body, resulting in the production of stress and immune-related proteins and a decrease in the energy allocated for growth, ultimately leading to an increase in energy consumption and a decrease in growth rate.	[128]

In summary, MP exposure will produce severe and complex toxic effects in animals. Because of their high specific surface area and non-degradable properties, MPs can lead to cellular oxidative stress and metabolic disorders, which finally lead to cell death. In addition, as nanoparticles, MPs can be enriched in different tissues and organs of the body, which then affect the immune, nervous, and reproductive systems. As carriers, MPs will adsorb and release various chemical substances and microorganisms, which indirectly endanger the health of animals. Although some progress has been achieved in the study of the mechanism of MP toxicity, the overall health effects and long-term consequences of MP exposure in animals are still unclear. Therefore, it is important to identify the exposure route of MPs and the mechanism of harmful effects on animal health to prevent and control the occurrence of related hazards. In production, livestock and poultry are inevitably exposed to MPs, and current studies mainly focus on monogastric animals. In fact, the feeding behavior and living environment of ruminants make them more likely to be exposed to MPs. Due to the large volume of the rumen, as well as the role of ruminal microorganisms in ruminants, foreign matter that enters the ruminants can be temporarily accommodated, thus giving an illusion of “strong tolerance” of ruminants to MPs. However, the MPs are still a serious threat to the health of ruminants.

## 5. Primary Pathways of MPs Entering Ruminants

After accumulation, degradation, and migration, MPs are ubiquitous in the environment. In the natural environment, MPs can inevitably enter ruminants through various exposure routes, such as ingestion, inhalation, and dermal contact, which threaten the life and health of ruminants and restrain the development of the ruminant industry (Figure 5A). This exposure is an important cause of potential and long-term harm to ruminant health.

### 5.1. Ingestion

In the feed industry, plastic products are often used to make engineering materials and feed packaging, which indirectly leads to the potential pollution of MPs in ruminant feed [132]. Feed is the basis for formulating diets for ruminants and usually covered and preserved with plastic film. Plastic mulching is widely carried out in agricultural production to improve crop yields, which is mainly utilized to control weeds, regulate soil temperature, and increase water use efficiency. Therefore, the extensive use of plastic products increases the potential risk of MP ingestion by ruminants [133]. In recent years, plastic mulching has been shown to be a major way for MPs to enter the livestock industry [134]. Moreover, hay bales are often wrapped in hemp rope or plastic netting to maintain the shape for better transport, both of which increase the use of plastic products in animals feed preservation. As a result, the potential risk of MPs or additives, such as phthalates and bisphenol A (BPA), being transferred from feed packaging to feed will be increased. Wang et al.’s study found that the bisphenol products (BP) were present in animal feed, and firstly demonstrated that BPA can migrate from plastic packaging to solid feed, with which the duration of exposure and initial BP concentration have a certain effects on the degree of migration [135]. Another study found that BP in PP and PE packaging materials can transfer to the solid feed of dairy cows, and may be transferred to milk, thereby endangering human health [136].

The plastic particles of agricultural soils come from a variety of sources, such as materials from greenhouses or tunnels, the use of silage film and plastic mulch, organic fertilizers, and sewage sludge [137]. In addition, tire fragments from agricultural mechanization are likely to contaminate the soil, and this plastic waste eventually accumulates in the soil in the form of MPs/NPs [138]. Plant roots easily absorb small plastic particles from the soil and transport them to edible parts through xylem pathways [139]. During grazing, it is very easy for farm animals to take MPs orally through grass or surface fresh water [140]. MPs are not only present in different feeds and grasses, but may also be present in water being contained in plastic tanks. Ruminants can ingest MPs by eating these feeds and grasses, as well as drinking water. In five grasslands, with a total survey area of 139,050 m^2^, in northern Bavaria of Germany, Meyer et al. determined the artificial debris in the stomach contents of 100 cattle and 50 sheep, and results showed that 30% of cattle and 6% of sheep contained plastic in their digestive tracts, the most abundant of which were the plastic sheets used to cover silage and the mesh fibers used to wrap hay bales [141]. Recently, in a study from Spain, soil samples from fields where plastic mulch had been used for at least a decade and fecal samples from sheep grazed after harvesting crops in the same area were analyzed to assess plastic pollution in Spanish agricultural soils and the ingestion of plastic by sheep, and results showed that MPs were detected in all sampled sheep, including animals that were not grazing directly on covered arable land [38]. MPs in animal excreta may return to the environment due to the livestock contamination, and MPs accumulate in water sources around the farm, which may be consumed by animals again, thus forming a vicious cycle.

### 5.2. Inhalation

The second way for ruminants to be exposed to MPs is through inhalation. Airborne MP transmission through the respiratory tract or oral cavity is an underrated potential route of entry into animals, and compared with MPs in food and water, airborne plastic particles can be directly inhaled into the lungs [142]. These MPs that enter the respiratory tract will likely come out by sneezing or coughing and enter the digestive system by swallowing. A recent study found that MP concentration in rainwater was higher than that in the discharge water of a sewage treatment plant, indicating that MPs in freshwater systems may originate from the air [143]. Therefore, MPs in the air can be returned to the soil and water through rainfall or snowfall. Airborne MPs are usually found in suspended particles or dust, which are likely to come from urban dust and paint- and oil-related industries. With the flow of the atmosphere, the transmission distance of MPs in the atmosphere can reach up to 95 km [144], and the MPs in the air are approximately 9.80/m^3^ [145]. There are a variety of plastic polymers in the air, including PP, PE, PET, and PS, most of which form fibrous particles [146]. A recent study has shown that airborne MPs can enter and accumulate in the lungs through respiration, causing lasting and deleterious effects in animals [147]. In livestock production, a large number of particles will enter the air. If the ventilation of the livestock house is insufficient, these particles can accumulate in the air, resulting in the reduction in air quality in the livestock house, which greatly increases the exposure of ruminants to airborne MPs [148]. When the humidity and air temperature in the barn are high, the air will contain a large number of microbial contaminants [149]. Due to the large surface area of MPs, they are allowed to form polymers with airborne microorganisms, which provides a relatively stable environment for microbial growth and reproduction, causing pathogenic microorganism exposure [129]. The feeding methods of ruminants can be divided into house, semi-house feeding, and grazing feeding. In addition to some areas with rich grassland resources, other areas are mainly house and semi-house feeding. Thus, ruminants are likely to be affected by the air quality in the barn and inhale MPs.

### 5.3. Dermal Contact

More and more researchers have paid attention to the impact of air pollutants on health. According to relevant statistics, billions of animals die every year [150]. MPs are ubiquitous in the environment due to their persistence and low-density characteristics, as well as inefficient waste management [151,152]. In recent years, researchers have found that MPs that were inhaled or ingested were able to be transported to the circulatory system and other organs, thereby causing harm to the body [131,144,153] (Figure 5C,D). In addition to ingestion and inhalation, dermal contact is also a potential route for MPs to enter ruminants. In barns, cooling MPs sprayed in air or water may directly contact with the animals’ skin [129]. The abundant hairs on the surface of the animal’s body can protect the skin to reduce the impact of MPs on the skin surface. However, if MPs contain or adhere to high lipophilic organics on the surface, it is easier for them to enter the skin. Additionally, the surface of the skin is not completely covered by the stratum corneum. Therefore, skin lesions, sweat glands, and hair follicles provide potential pathways for MP invasion. The skin of ruminants also has a certain adsorption capacity, and MPs may attach to the skin surface after contact with the skin. One study has found that MPs with specific chemical structures or a charged surface are more likely to attach to the skin surface [154]. During daily activities, ruminants may experience skin damage such as scratches and bites, and, hence, MPs can enter the animals through these wounds. When the skin is damaged, the skin barrier function at the wound site is impaired, and MPs are more likely to penetrate the skin tissue into the lymphatic system or blood circulation [155]. Up to now, research on the entrance of MPs into ruminants through dermal contact remains scarce, and more studies mainly focus on MP ingestion. With the in-depth research of MP contamination, more studies related to skin exposure to MPs are still needed in animal and human studies.

## 6. Impact of MPs on Ruminants

Plastic products are commonly used in feed packaging, as well as engineering materials, which indirectly contributes to the potential threat of MPs in ruminant feed. MP exposure will produce a toxic response to the body and then cause harm to animals. Related studies have found the presence of MPs in ruminants’ bodies and feces [134,156,157,158]. Therefore, it is important to explore the harm of MPs on ruminants. Existing studies have proved that MP exposure in ruminants can affect production performance, immune function, and rumen microbial diversity.

### 6.1. Production Performance

In most developing countries, ruminants are often grazed freely in open pasturing areas, where they indiscriminately ingest plastic foreign matter. Indigestible plastic products accumulate in the rumen of ruminants, resulting in rumen impaction, recurrent tympanic cavity, indigestion, and many other adverse health effects [159]. Due to the ingestion of plastic products, rumen impaction has been observed in all domestic and wild ruminants. Among all domestic ruminants, compared to sheep, cattle are more likely to undergo rumen impaction because of plastic materials, which may be attributed to the feeding habits of cattle. Since cattle graze close to the ground, they have a low ability to distinguish food carefully in the process of intake. Sheep have flexible lips, as well as selective eating behaviors, and, thus, it is not easy for sheep to ingest plastic foreign matter. However, environmental pollution, feed shortage, and reduced animal feeding standards will also force sheep to ingest plastic products [160]. Plastics ingested into the rumen cause the slow release of chemicals in the ruminal fluid, which can enter the food chain through milk and meat products [161]. After MPs enter the intestine, they may accumulate in the narrow part of the intestine. The intestines of ruminants are long and curved, and small-sized MPs will mix with food residues in the intestines to form larger polymers, which may induce intestinal obstruction, hinder the normal transportation of intestinal contents, and cause constipation and abdominal pain in animals. In addition, this adverse effect may even lead to life-threatening diseases such as peritonitis and intestinal perforation, thereby hindering the normal growth of ruminants.

Due to their large surface area, MPs are prone to absorb toxic chemicals, including the plasticizers and heavy metals used in the manufacture of plastics, which may be desorbed and released from the surface of MPs under the action of the digestive system, causing dysfunction of digestive system of ruminants [157]. Plastic materials have been found to irritate the rumen mucosa, leading to ruminitis and hindering fermentation by interfering with the movement of the rumen contents [162]. At the same time, ingested MPs may cause digestive system disorders in ruminants, thus affecting their normal growth [163]. A previous study found that the ingestion of MPs in sheep may cause digestive disorders, including intestinal obstruction and recurrent rumen tympanite, which prevent their normal growth [38]. In addition, a recent study has demonstrated that MP ingestion reduced the digestibility of dry matter, fiber, and lipids, thus decreasing the daily gain of lambs. MPs can also disturb intestinal movement and then affect the digestion and absorption of nutrients (Figure 5B). Lambs’ ingestion of MPs reduced the quality and nutritional value of meat, and these effects finally decreased the production performance of lambs [130].

### 6.2. Immune Function

Oxidative stress is the main cause of the onset of various inflammatory diseases, and the cell membranes of immune cells are rich in unsaturated fatty acids, which makes them more sensitive to ROS. When immune cells are damaged, they interfere with the secretion of antibodies and cytokines and the presentation of antigens, which leads to immune dysfunction and increases the risk of disease infection [164]. Under normal circumstances, the immune system of ruminants is in a state of homeostasis, which effectively removes pathogens and allows the body to maintain a normal state. The entrance of MPs into ruminants will break this balance, which is attributed to the fact that MPs will lead to the occurrence of inflammation and the change in immune cell function, causing the insufficient immune response of the body. Wang et al. investigated the cytotoxicity of PS MPs on goat mammary epithelial cells (GMECs). The results showed that, compared with the control group, PS MP treatment significantly reduced cell viability, damaged organelle integrity, and changed cell morphology. The detection of membrane sites and ROS indicated that PS MPs could induce mitochondrial dysfunction and oxidative stress. Moreover, PS MPs induced endoplasmic reticulum stress via the PERK/elF2α/CHOP pathway and are associated with intracellular Ca^2+^ overload, and the addition of PERK inhibitors (ISRIB) attenuated PS-MPS-induced endoplasmic reticulum stress and apoptosis, which suggested that endoplasmic reticulum stress may exacerbate cytotoxicity induced by PS MPs [165]. On the other hand, MPs’ surfaces may adsorb some pollutants, such as heavy metals and organic matter, which can stimulate the immune system after release in the body. As an important immune organ of ruminants, the spleen plays a key role in the immune response, as well as in the storage and activation of immune cells. Long-term MP exposure may lead to structural changes in spleen tissue. Unfortunately, no relevant studies have been reported. In the future, more animal experiments should be conducted to explore the effects of MP ingestion on spleen function.

### 6.3. Rumen Microbiota

The microbial community in the rumen is crucial to the health, growth and development, digestion, and absorption of ruminants, and it plays a key role in promoting the absorption of nutrients, maintaining the homeostasis of the host’s digestive system, and regulating immune function [166]. The rumen’s microbial community is composed of a variety of microorganisms, such as bacteria, fungi, archaea, and protozoa, which interact and compete with each other [167]. The microorganisms form a complex microbial ecosystem in the rumen that affects the physiological function of the host, and one study has shown that rumen microbial dysregulation is closely related to the occurrence of a variety of diseases, including diarrhea, rumen inflammation, and metabolic diseases [168]. In recent years, the harm of MPs to the ecological environment, as well as to humans and animals, has become increasingly apparent. Relevant researchers have studied the effects of MPs on microorganisms in the rumen of ruminants, but this kind of research is still in its infancy, and the information is limited. Some studies have found that MPs reduced the abundance of some bacteria (e.g., *Prevotellaceae_YAB2003_group* and *Coriobacteriales_Incertae_Sedis*) in the rumen, which play a key role in ruminant digestion, while they increased the relative abundance of pathogenic bacteria, including actinomycetes and Proteobacteria. Previously, an experiment in lambs found that MP exposure mainly reduced ammonia nitrogen in the rumen and did not obviously affect the diversity of ruminal microorganisms. However, the relative abundance of some microorganisms was changed, induced by MP exposure, which was reflected in the increased relative abundance of Bacteroidetes and actinomycetes and the decreased relative abundance of bacteria related to cellulose degradation and aromatic amino acid synthesis. In addition, MPs also caused abnormal blood physiology and metabolism, oxidative stress, and liver and kidney swelling in lambs [130].

The biggest harm of plastics to the environment lies in their degradation-resistant ability, and researchers have explored a lot of solutions for plastic degradation, but few results have achieved. Ruminant diets may contain natural plant polyester (keratin), and enzymes that hydrolyze these substances may exist in the rumen. Thus, the rumen of ruminants may have the ability to degrade plastics [169]. Quartinello et al. used the ruminal fluid of dairy cows to conduct a hydrolysis study on three synthetic polyesters, namely PET, polybutylene adipate/terylene phthalate (PBAT), and polyethylene 2,5-furandicarboxylate (PEF), and the results showed that the combined action of bacteria, archaea, and eukaryotes in the ruminal fluid could hydrolyze these synthetic polyesters [170]. Based on this, whether ruminal microorganisms can degrade MPs is worthy of further study. In addition, MPs may reduce the fermentation function of ruminal microorganisms. This is because MP ingestion may adsorb nutrients in the rumen and reduce the substrate available to microorganisms, resulting in a decrease in volatile fatty acid production, which not only affects the fermentation function of microorganisms, but also affects the energy intake of ruminants. The effects of MPs on the other digestive organs of ruminants have not received enough attention, and in-depth studies are needed in the future.

## 7. Conclusions and Future Viewpoints

MPs have a direct threat of toxicity to the animal body and are carriers of chemical poisons and pathogens that endanger animal health and seriously affect animal husbandry and the production of animal-origin foods. The toxic mechanism of MPs is relatively complex, mainly including oxidative stress, immunotoxicity, reproductive toxicity, and neurotoxicity. The toxicity level is closely related to the type, exposure dose, and size and shape of MPs. MPs can enter the bodies of ruminants through ingestion, inhalation, and skin contact, thus affecting their health and production performance. Among them, the research on ruminants’ exposure to MPs mainly focuses on ingestion and its related negative effects. Due to the inevitable long-term exposure, the harm of MPs in ruminants is mainly manifested in the effects on production performance, immune function, and microbial community in the rumen.

In summary, the research on the toxicity mechanism of MPs and the harm caused to ruminants is still insufficient, and there are some shortcomings. Moreover, the problem of MP pollution still needs to be further studied. For example, whether plastic foreign materials ingested by ruminants can be degraded to MPs by microorganisms has not been demonstrated. The toxicology of MPs has been studied mainly in various cells and rodent models, but toxicity studies in some terrestrial mammals (e.g., pigs, cattle, and sheep) are often neglected. Studies on the toxicology of MPs are still lacking, and, in particular, there is no mature research method to investigate the adverse effects on animals exposed to high MP concentrations for a long time. Although MPs have been found in all types of livestock ecosystems, the exact pathways of MP entry and nutrient transfer have not been fully studied. The lack of internationally recognized standardized methods for sampling and validation is considered a major limitation in MP research. Whether MPs in lactating animals can be transferred to milk through the mammary glands needs further study. In the future, multi-omics technology and in vitro experiments should be used to study the microbial community in the rumen to better guide the production of ruminants. In addition, as the impact of MPs continues to increase, developing sustainable solutions to mitigate their harmful effects and reduce their presence in the environment is essential for the survival of humans and animals, as well as for the development of the livestock industry. At the same time, future research still needs to continuously study the physical and chemical effects of MPs on the digestive organs of ruminants, such as the rumen and intestine, as well as their distribution patterns. It is crucial to reveal the toxic mechanisms of MPs in ruminants, including bioaccumulation effects and chemical toxicity. Additionally, research should focus on the residual presence of MPs in ruminant products and their potential risks to human health.

## Figures and Tables

**Figure 3 biomolecules-15-00462-f003:**
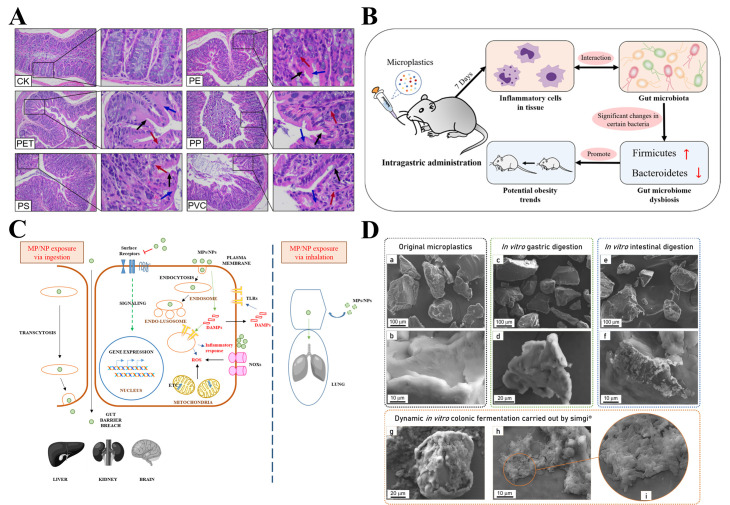
Effects of MP exposure on mice, potential cytotoxic mechanisms of MPs/NPs, and scanning electron microscopy of PET MPs. (**A**) Effects of MPs on colon histopathology. Representative images of colon tissue H&E (200×). Red arrow: inflammatory cells. Black arrow: crypt surface irregular. Blue arrow: reduced goblet cells [71]. (**B**) Effects of MP exposure on mice [71]. (**C**) Schematic of the underlying cellular mechanisms of MP/NP toxicity. MPs/NPs can be absorbed by ingestion and inhalation. MPs/NPs can damage the plasma membrane and the intestinal barrier (left). These may also disrupt signaling at cell surface receptors and alter gene expression in the nucleus. Endocytic MPs/NPs also disrupt endocytic pathway function and damage endosomal membranes. The above stress activates the innate immune system, and endogenous- and secretory-damage-associated molecular patterns (DAMP) induce innate immune-mediated toll-like receptors (TLRs). Stress can induce NADP oxidase (NOXs) to produce ROS. Mitochondrial damage, either MPs/NPs from the endosome or stress responses, can produce more ROS by impinging on the efficiency of the electron transport chain (ETC) process. If the intestinal vascular barrier is damaged, MPs/NPs can enter circulation, or they may occur through endocytosis and, thus, reach other organs. The lungs may have a more direct route to exposure to MPs/NPs in the air (right) [74]. (**D**) Field Emission Scanning Electron Microscopy (FESEM) micrographs of PET MPs: (**a**,**b**) raw PET MPs, (**c**,**d**) PET MPs after in vitro gastric digestion, (**e**,**f**) PET MPs after in vitro gastrointestinal digestion, and (**g**–**i**) PET MPs after in vitro gastrointestinal digestion and colonic fermentation [72].

**Figure 5 biomolecules-15-00462-f005:**
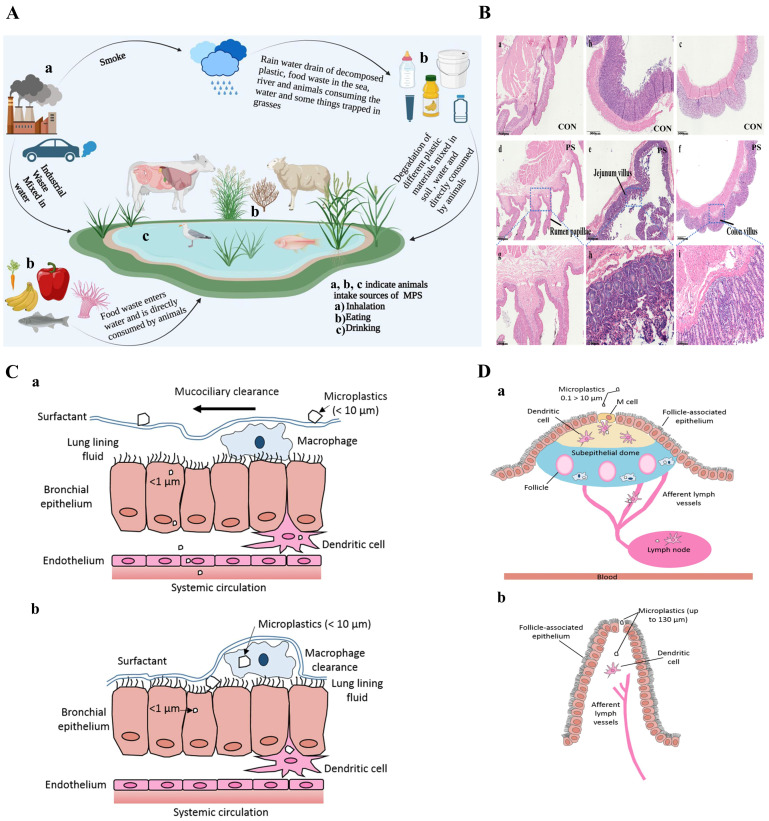
Schematic diagram of the pathways of animal ingestion of MPs and the effects of MPs on the gastrointestinal tract. (**A**) Pathways of MP intake by animals [129]. (**B**) PS gastrointestinal section before and after exposure: H&E images of the rumen (**a**), jejunum (**b**), and colon (**c**) in the CON group; (**d**) rumen H&E map of S-PS group; (**e**) L-PS group jejunum H&E map; (**f**) colon H&E map of L-PS group; and (**g**–**i**) are locally enlarged images corresponding to (**d**–**f**), respectively [130]. (**C**) Potential uptake and clearance mechanisms of MPs (0.1 > 10 μm) in the lungs: (**a**) There is a reduced chance of lung fluids (surfactants and mucus) causing microplastic displacement in the upper respiratory tract (central lung) where the lung membrane is thicker. The ciliary mucosa with particles >1 μm was cleared. Particles <1 μm can be ingested through the epithelium. (**b**) If the aerodynamic diameter of the microplastic allows deposition deep in the lung, it may penetrate the thinner lung fluid and contact the epithelium, metastasizing through diffusion or active cell uptake [131]. (**D**) Predicted pathways for uptake of MPs from the gastrointestinal tract (GIT): (**a**) MPs uptake from GIT lumen by M cells of Pyle’s collecting lymph nodes through endocytosis (0.1 > 10 μm). M cells sample and transport particles from the intestinal lumen to mucosal lymphoid tissue. (**b**) MPs are absorbed from the GIT lumen by paracellular absorption. Non-degradable particles, such as MPs, can be mechanically pinched into the underlying tissue through loose connections in the single-celled upper layer. Dendritic cells are able to devour these particles, transporting them to lymph vessels and veins below. They may be distributed to secondary tissues, including the liver, muscles, and brain [131].

## Data Availability

No new data were generated during the preparation of this study. All quoted figures, facts and conclusions are based on published information and literature. Because this study did not involve direct research on individuals or sensitive information, no data sets specific to this study were collected or generated. In accordance with MDPI’s research data policy, we hereby state that no data sets directly related to this study are available for sharing.

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
