# Peer review of "Toxicity Mechanisms of Microplastic and Its Effects on Ruminant Production: A Review"

_biomolecules, 2025, doi:10.3390/biom15040462_

Round 1
Reviewer 1 Report
Comments and Suggestions for Authors
Dear Authors,
The manuscript is well designed and the literature is prepared well.
Please correct the resolution of the graphics.
I don't have more corrections to this manuscript.
Author Response
Dear Reviewer,
On behalf of all co-authors, I would like to appreciate the reviewer for the positive and constructive comments and suggestions concerning our manuscript. We have gone through all the comments in detail and have tried our best to revise our manuscript according to the comments. Below we provide a copy of the comments with our point-to-point responses.
Once again, we would like to express our great appreciation to you for comments on our manuscript. Looking forward to hearing from you.
With best regards,
Corresponding Author
Guangdong Ocean University, China
majian0411@gdou.edu.cn
Response to Reviewer 1 Comment
Comment 1: The manuscript is well designed and the literature is prepared well. Please correct the resolution of the graphics. I don't have more corrections to this manuscript.
Response 1: Thank you for pointing this out. We have improved the resolution of the graphics in the revised manuscript.
Reviewer 2 Report
Comments and Suggestions for Authors
Dear authors,
Let me start by congratulating you on the article. Overall, the manuscript is complex and contain new valuable and interesting information.
Chapter 1 - Introduction:
Strengths: These section is well-structured and provides a logical introduction, with significant illustration (Figure 1), to the topics covered in the review. I appreciated that the introduction explicitly outlined the key chapters and aspects to be covered next.
In the introduction, I did not identify any weaknesses that would require changes.
Chapter 2 - Methods
Strengths: I appreciated that this review clearly specifies the sources utilized. The authors mention that the literature search was conducted between 2010 and 2024 across databases such as Web of 129 Science, Science Direct, PubMed, Scopus and Google search engines, focusing on articles relevant to the topic. The search employed specific keywords, including [list the keywords].
Weaknesses: Additionally, it would be beneficial to clarify the inclusion and exclusion criteria applied during the selection of articles, beyond just the publication year, as this would enhance the transparency of the methodology.
Chapter 3 - Types of microplastics
Strengths: Aside from the weaknesses noted below, I believe this chapter is well-structured, professionally written, and contains valuable information.
Weaknesses:
- I would restructure Figure 2 in a more logical manner. I suggest starting with: (C) Environmental behavior of organic fertilizer MPs in soil, (D) Possible sources of MPs in livestock breeding, and (E) The fate of MPs linked to the disposal of livestock and poultry waste. After this section, we can move on to the observed effects in laboratory animals: (A) MPs accumulation of different sizes in mouse tissues after 28 days of exposure and (B) Representative images of H&E-stained liver sections of mice exposed to 0.5 mg/d PS-MPs (5 μm and 20 μm) for 4 weeks (black arrows indicating liver inflammation, white arrows showing fat droplets). I would even suggest dividing this figure into two separate figures for better clarity (Especially since figure 3 shows the effects of various types of MPs in mice). Additionally, in the text (lines 145-147) they begin with Figure 2C.
- Table 1 (mentioned on line 143) should be inserted in the text before the figure or before subchapter 3.1.
Chapter 4 - Toxicity mechanism of microplastics
Strengths: In general, the chapter is well structured, but certain changes (mentioned below) would increase its quality.
Weaknesses:
- I believe Table 2 (Summary of Toxic Effects and Mechanisms of MPs) should be restructured to clearly align with the mechanisms discussed in the text (e.g., Oxidative Stress, Immunotoxicity, Reproductive Toxicity, etc.). Each mechanism should have dedicated rows listing the studied species. In its current form, the table is difficult to follow. The information is valuable, and a clearer structure would enhance the article’s quality and readability. Also, this Table (mentioned on line 327) should be inserted in the text before the figure or before subchapter 4.1.
- Considering the data presented in Subchapter 4.6 (Deoxyribonucleic Acid Damage), I believe it should be included in the first section, under the mechanism related to oxidative stress.
Chapter 5 - Main ways of MPs to enter ruminants
Strengths: Overall, the chapter is well-structured, but the suggested changes (listed below) would enhance its quality.
Weaknesses:
- It's not a negative observation per se, but I would support the following title for this chapter - Primary Pathways of MPs Entering Ruminants
- The title of Figure 4 (Schematic diagram of PS MPs accumulation in the lung and their effects on the gastrointestinal tract, and predicted pathways for uptake of MPs from the gastrointestinal tract and the lung) is too long, especially considering the subsequent description of the subsections. Additionally, the data are derived from experimental studies on mice. Personally, I believe this figure does not belong in this chapter.
Chapter 6 - Harm of MPs to ruminants
Strengths: I believe this chapter is well-structured and provides valuable information.
Weaknesses: This isn't necessarily a negative observation, but I would recommend the following title for this chapter: Impact of MPs on Ruminants.
In conclusion, I believe the structure of this review should be re-evaluated to present the data—which is both interesting and valuable—in a reader-friendly manner. This would enhance clarity and make it easier for the audience to fully understand the content.
Author Response
Dear Reviewer,
On behalf of all co-authors, I would like to appreciate the reviewer for the positive and constructive comments and suggestions concerning our manuscript. We have gone through all the comments in detail and have tried our best to revise our manuscript according to the comments. Below we provide a copy of the comments with our point-to-point responses.
Once again, we would like to express our great appreciation to you for comments on our manuscript. Looking forward to hearing from you.
With best regards,
Corresponding Author
Guangdong Ocean University, China
majian0411@gdou.edu.cn
Detailed responses to reviewer,
All comments provided by reviewer are given in gray italics, and our responses are in black non-italicized. Additionally, revised portion are marked in green in the revised manuscript.
Response to Reviewer 2 Comments
Revised portions are marked in green in the revised manuscript.
Comment 1: In the introduction, I did not identify any weaknesses that would require changes.
Response 1: Thanks for your recognition of our work.
Comment 2: Additionally, it would be beneficial to clarify the inclusion and exclusion criteria applied during the selection of articles, beyond just the publication year, as this would enhance the transparency of the methodology.
Response 2: Thank you for pointing this out. We agree with this comment. Therefore, we added relevant information in the methods section (page 5; Line 152-159).
Comment 3: I would restructure Figure 2 in a more logical manner. I suggest starting with: (C) Environmental behavior of organic fertilizer MPs in soil, (D) Possible sources of MPs in livestock breeding, and (E) The fate of MPs linked to the disposal of livestock and poultry waste. After this section, we can move on to the observed effects in laboratory animals: (A) MPs accumulation of different sizes in mouse tissues after 28 days of exposure and (B) Representative images of H&E-stained liver sections of mice exposed to 0.5 mg/d PS-MPs (5 μm and 20 μm) for 4 weeks (black arrows indicating liver inflammation, white arrows showing fat droplets). I would even suggest dividing this figure into two separate figures for better clarity (Especially since figure 3 shows the effects of various types of MPs in mice). Additionally, in the text (lines 145-147) they begin with Figure 2C.
Response 3: Thanks for your valuable advices. We believe that this will greatly improve the quality of our manuscript. Thus, we have reassembled Figure 2 in the revised manuscript according to your suggestion (page 8; Figure 2).
Comment 4: Table 1 (mentioned on line 143) should be inserted in the text before the figure or before subchapter 3.1.
Response 4: Thank you for pointing this out. We have inserted Table 1 in the text before Figure 2 (page 5-7; Table 1).
Comment 5: I believe Table 2 (Summary of Toxic Effects and Mechanisms of MPs) should be restructured to clearly align with the mechanisms discussed in the text (e.g., Oxidative Stress, Immunotoxicity, Reproductive Toxicity, etc.). Each mechanism should have dedicated rows listing the studied species. In its current form, the table is difficult to follow. The information is valuable, and a clearer structure would enhance the article’s quality and readability. Also, this Table (mentioned on line 327) should be inserted in the text before the figure or before subchapter 4.1.
Response 5: Thank you very much for your detailed review. Based on your suggestions, we have restructured the Table 2 (page 13-15; Table 2).
Comment 6: Considering the data presented in Subchapter 4.6 (Deoxyribonucleic Acid Damage), I believe it should be included in the first section, under the mechanism related to oxidative stress.
Response 6: Thank you for pointing this out. We agree with this comment. Thus, we have included Subchapter 4.6 (Deoxyribonucleic Acid Damage) in 4.1 Oxidative stress in the revised manuscript (page 16; Line 381-400).
Comment 7: It's not a negative observation per se, but I would support the following title for this chapter - Primary Pathways of MPs Entering Ruminants.
Response 7: Thanks for your valuable comment. Under your advice, we have changed the title of Part 5 to Primary pathways of MPs entering ruminants (page 21; Line 620).
Comment 8: The title of Figure 4 (Schematic diagram of PS MPs accumulation in the lung and their effects on the gastrointestinal tract, and predicted pathways for uptake of MPs from the gastrointestinal tract and the lung) is too long, especially considering the subsequent description of the subsections. Additionally, the data are derived from experimental studies on mice. Personally, I believe this figure does not belong in this chapter.
Response 8: Thanks for your good proposal and we believe that this will greatly improve the quality of the manuscript. We have modified the notes in Figure 5 and Figure 5A (page 24; Figure 5).
Comment 9: This isn't necessarily a negative observation, but I would recommend the following title for this chapter: Impact of MPs on Ruminants.
Response 9: Thanks for your valuable comment. Under your advice, we have changed the title of Part 6 to Impact of MPs on ruminants (page 25; Line 742).
Reviewer 3 Report
Comments and Suggestions for Authors
The manuscript titled “Toxicity mechanisms of microplastic and its effects on ruminants production: A review” by Su, M.; et al. is a Review work where the authors outlined the most recent advances in the detrimental effects of the microplastic intake in ruminants that is crucial to keep the quality and productivity of the farming Industry. The manuscript is generally well-written and this is a topic of growing interest.
However, it exists some points that need to be addressed (please, see them below detailed point-by-point) to improve the scientific quality of the submitted manuscript paper before this article will be consider for its publication in Biomolecules.
1) Introduction. “With the development (…) approximate 400 million tons of plastic materials are produced globally each year (…) the other plastic waste remains in the environment” (lines 28-42). Could the authors provide quantitative data insights according to the worldwide economic impact to recycle plastic materials? This will significantly aid the potential readers to better understand the significance of this Review work.
2) “The MPs are widely found in soil, drinking water, air and other environmental media (…)” and animals” (lines 50-52). Here, it should be also discussed abot the existing detection techniques of microplastics. In this framework, biomonitoring tools [1] or single-molecule techniques [2] needs to be mentioned among other approaches.
[1] https://doi.org/10.1016/j.envadv.2025.100609
[2] https://doi.org/10.22034/IAR.2022.1965012.1317
3) Finally, the authors also are requested to claim the novelty of this work compared to other published Review works in this field. A brief statement should be added in this regard.
3) 2. Methods. This section is clearly explained. No actions are requested from the authors.
4) 3. Types of microplastics. What is the toxic concentration threshold linked to each examined microplastic? Eventually this information could be furnished in the Table 1 (lines 317).
5) 4. Toxicity mechanisms of microplastics. A schematic representation related to the underlying toxic mechanisms leaded to the microplastic uptake will benefit the potential readers.
6) 5. Main ways of MPs to enter ruminants. This section is clearly detailed. No actions are requested from the authors.
7) 6. Harm of MPs to ruminants. Some insights about the hepatotoxicity caused the accumulation of MPs in the liver based on the size and dimensions should be depicted in this subsection.
8) “7. Conclusions and future viewpoints (lines 812-841). This section perfectly remarks the most relevant outcomes found by the authors in this field and also the promising future perspectives. It would be desirable to add a brief statement to state the potential future action lines to pursue the topic covered in this Review work.
Author Response
Dear Reviewer,
On behalf of all co-authors, I would like to appreciate the reviewer for the positive and constructive comments and suggestions concerning our manuscript. We have gone through all the comments in detail and have tried our best to revise our manuscript according to the comments. Below we provide a copy of the comments with our point-to-point responses.
Once again, we would like to express our great appreciation to you for comments on our manuscript. Looking forward to hearing from you.
With best regards,
Corresponding Author
Guangdong Ocean University, China
majian0411@gdou.edu.cn
Detailed responses to reviewer,
All comments provided by reviewer are given in gray italics, and our responses are in black non-italicized. Additionally, revised portion are marked in yellow in the revised manuscript.
Response to Reviewer 3 Comments
Revised portions are marked in yellow in the revised manuscript.
Comment 1: Introduction. “With the development (…) approximate 400 million tons of plastic materials are produced globally each year (…) the other plastic waste remains in the environment” (lines 28-42). Could the authors provide quantitative data insights according to the worldwide economic impact to recycle plastic materials? This will significantly aid the potential readers to better understand the significance of this Review work.
Response 1: Thanks for your good proposal and we believe that this will greatly improve the quality of the manuscript. We have added quantitative data insights on the impact of plastic material recycling on the global economy in the revised manuscript (page 2; line 42-47).
Comment 2: “The MPs are widely found in soil, drinking water, air and other environmental media (…)” and animals” (lines 50-52). Here, it should be also discussed abot the existing detection techniques of microplastics. In this framework, biomonitoring tools [1] or single-molecule techniques [2] needs to be mentioned among other approaches.
[1] https://doi.org/10.1016/j.envadv.2025.100609
[2] https://doi.org/10.22034/IAR.2022.1965012.1317
Response 2: Thanks for your valuable comments. We agree with this comment. Therefore, we have added the relevant information of the existing microplastic detection technology in the revised manuscript (page 2; line 58-69).
Comment 3: Finally, the authors also are requested to claim the novelty of this work compared to other published Review works in this field. A brief statement should be added in this regard.
Response 3: Thank you for pointing this out. We agree with this comment. Therefore, we have added relevant information about the novelty of our paper compared to other published reviews in the field (page 3; line 131-136).
Comment 4: 2. Methods. This section is clearly explained. No actions are requested from the authors.
Response 4: Thanks for your recognition of our work. In order to further improve the manuscript, we added some content in the 2. Methods according to the comments of another reviewer to enhance the transparency of the methodology.
Comment 5: 3. Types of microplastics. What is the toxic concentration threshold linked to each examined microplastic? Eventually this information could be furnished in the Table 1 (lines 317).
Response 5: Thank you for pointing this out. I'm sorry to say that the research on the toxicity concentration thresholds of MPs is still evolving. Due to the complexity of factors such as the types, shapes, sizes, chemical compositions and environmental conditions of microplastics, as well as the varying sensitivities of different organisms to MPs, determining a unified toxicity concentration threshold is challenging. Currently, research on the toxicity concentration thresholds of MPs primarily focuses on aquatic animals. For example, Zhou et al. exposed Litopenaeus vannamei to water containing 0.1, 1.0, 5.0, and 20 μm PS MPs at concentrations of 10 mg/L and 1 mg/L for 24 hours and 12 days to investigate the size-dependent toxicity of PS MPs. The results showed that after 24 hours of acute exposure, the ingestion and excretion of MPs were size-dependent, with smaller particles exhibiting higher bioavailability. After 12 days of subchronic exposure, the smaller size of the MPs, the greater the damage to the viscera and gills, indicating that PS MPs exposure causes size-dependent damage to shrimp. Additionally, the impact of specific particle sizes on certain research indicators may vary [1].
[1] DOI: https://doi.org/10.1016/j.envpol.2022.120635
Comment 6: 4. Toxicity mechanisms of microplastics. A schematic representation related to the underlying toxic mechanisms leaded to the microplastic uptake will benefit the potential readers.
Response 6: Thank you very much for your good proposal and we believe that this will greatly improve the quality of the manuscript. Thus, we have added Figure 4. Schematic of cytotoxicity, immunotoxicity and neurotoxicity of MPs in the revised manuscript (page 19; Figure 4).
Comment 7: 5. Main ways of MPs to enter ruminants. This section is clearly detailed. No actions are requested from the authors.
Response 7: Thanks for your recognition of our work.
Comment 8: 6. Harm of MPs to ruminants. Some insights about the hepatotoxicity caused the accumulation of MPs in the liver based on the size and dimensions should be depicted in this subsection.
Response 8: Thanks for your valuable comments. Sorry to say that as far as current studies are concerned, no studies have revealed the accumulation of MPs in ruminants and the resulting hepatotoxicity based on the size and dimensions of MPs. The study mentioned in our paper, which found that PS MPs caused liver and kidney swelling in lambs, was based on slaughter indicators (the liver weight of lambs exposed to S-PS (25 μm PS MPs) was significantly higher than that of the control group, M-PS (50 μm PS MPs), and L-PS (100 μm PS MPs)). It is believed that this question can be solved with the continuous deepening of research [2]. Thanks for your careful review again.
[2] DOI: https://doi.org/10.1016/j.ecoenv.2024.116389
Comment 9: 7. Conclusions and future viewpoints (lines 812-841). This section perfectly remarks the most relevant outcomes found by the authors in this field and also the promising future perspectives. It would be desirable to add a brief statement to state the potential future action lines to pursue the topic covered in this Review work.
Response 9: Thanks for your recognition of our work. Under your advice, we have added about possible future directions of action in the revised manuscript (page 27-28; line 880-886).
Round 2
Reviewer 3 Report
Comments and Suggestions for Authors
The authots did a great deal of effort to cover all the raised suggestions by the Reviewers. For this reason, the scientific manuscript was greatly improved. Based on the significance of the scope, I warmly endorse this work for further publication in Biomolecules journal